# Improving the Recyclability of an Epoxy Resin through the Addition of New Biobased Vitrimer

**DOI:** 10.3390/polym15183737

**Published:** 2023-09-12

**Authors:** Antonio Veloso-Fernández, Leire Ruiz-Rubio, Imanol Yugueros, M. Isabel Moreno-Benítez, José Manuel Laza, José Luis Vilas-Vilela

**Affiliations:** 1Grupo de Química Macromolecular (LABQUIMAC), Departamento de Química Física, Facultad de Ciencia y Tecnología, Universidad del País Vasco UPV/EHU, 48940 Leioa, Spain; leire.ruiz@ehu.eus (L.R.-R.); and josemanuel.laza@ehu.eus (J.M.L.); joseluis.vilas@ehu.eus (J.L.V.-V.); 2BCMaterials, Basque Center for Materials, Applications and Nanostructures, UPV/EHU Science Park, 48940 Leioa, Spain; 3Grupo de Química Macromolecular (LABQUIMAC), Departamento de Química Orgánica e Inorgánica, Facultad de Ciencia y Tecnología, Universidad del País Vasco UPV/EHU, 48940 Leioa, Spain; mariaisabel.moreno@ehu.eus

**Keywords:** sustainable materials, epoxy resin, Schiff base, epoxidized soybean oil, epoxy vitrimer, reprocessability, recyclability

## Abstract

In recent decades, the use of thermoset epoxy resins (ER) has spread to countless applications due to their mechanical properties, heat resistance and stability. However, these ERs are neither biodegradable nor recyclable due to their permanent crosslinked networks and usually, they are synthesized from fossil and toxic precursors. Therefore, reducing its consumption is of vital importance to the environment. On the one hand, the solution to the recyclability problems of epoxy resins can be achieved through the use of vitrimers, which have thermoset properties and can be recycled as thermoplastic materials. On the other hand, vitrimers can be made from natural sources, reducing their toxicity. In this work, a sustainable epoxy vitrimer has been efficiently synthesized, VESOV, by curing epoxidized soybean oil (ESO) with a new vanillin-derived Schiff base (VSB) dynamic hardener, aliphatic diamine (1,4-butanediamine, BDA) and using 1,2-dimethylimidazole (DMI) as an accelerator. Likewise, using the same synthesized VSB agent, a commercial epoxy resin has also been cured and characterized as ESO. Finally, different percentages (30, 50 and 70 wt%) of the same ER have been included in the formulation of VESOV, demonstrating that only including 30 wt% of ER in the formulation is able to improve the thermo-mechanical properties, maintaining the VESOV’s inherent reprocessability or recyclability. In short, this is the first approach to achieve a new material that can be postulated in the future as a replacement for current commercial epoxy resins, although it still requires a minimum percentage of RE in the formulation, it makes it possible to recycle the material while maintaining good mechanical properties.

## 1. Introduction

In recent years, special attention has been paid to the use of epoxy resins (ER). This type of thermoset polymer has distinguished properties, such as thermal stability, mechanical strength, creep resistance, electrical insulation, and chemical resistance [1,2,3,4,5,6,7]. These polymers are industrially synthesized to use as coatings, adhesives, electronic packaging materials or composites for automobile, aerospace or transportation industries [1,2,8,9]. In fact, the global annual production in 2020 reached almost 10 million tons [10].

Nevertheless, most of the current epoxy thermosets (~90%) are prepared from non-renewable diglycidyl ether of bisphenol A (DGEBA) and cannot be reprocessed or recycled due to their permanent crosslinking, which causes significant waste and environmental problems after their service lifetime [11,12,13]. In addition to the non-renewability, bisphenol A and epichlorohydrin, which are the raw materials of DGEBA, are toxic, fossil derivatives and both are categorized as hazardous to living organisms [14].

Therefore, recently, more attention has been paid to designing sustainable epoxy thermoset resins from diverse renewable resources, such as epoxidized vegetable oils [15], cardanol [16], isosorbide [17], vanillin [18,19], etc. Specifically, vegetable oils are prime candidates to replace fossil-based derivatives in polymer materials due to: (i) their universal availability, (ii) low toxicity and (iii) low price [20]. Moreover, the presence of carbon-carbon double bonds enables them to be easily transformed into epoxidized vegetable oils (EVOs) through a curing process with hardening agents. Nevertheless, their highly crosslinked structure combined with slightly flexible backbones provides EVOs with poor mechanical strength fusing with poor ductility and low glass transition temperature (T_g_) [21,22].

To address these issues, new studies to develop new epoxy resins that contain the remarkable properties of thermosets, as well as the intrinsic capacity of thermoplastics to be recycled after their useful life, are needed. One of the possible solutions is the development of covalent adaptable networks (CANs).

The so-called CANs are polymeric materials with permanent crosslinks, which can reversibly transform into dynamic crosslinks, allowing their chains to flow (analogous to thermoplastics) when they are induced by an external stimulus, such as temperature [23,24], exposure to ultraviolet light [25] or pH [26,27]. In the absence of this stimulus, their reticulated structure offers them stiffness and durability as thermosetting materials. Thus, these polymeric materials, which have thermosetting polymer properties due to their crosslinking networks, are recyclable and reusable due to the dynamic nature of these crosslinks.

In 2011, Leibler et al. reported [28] a new class of CANs called vitrimers, which resemble vitreous silica due to their change in viscosity and the fact that they also follow an Arrhenius relationship that increases with temperature. Vitrimers belong to a sub-class of CANs in which the crosslinking bonds have an associative nature, resulting in the ability of the material to change its topology via exchange reactions [29,30,31,32,33]. Certainly, vitrimer crosslink density can be recognized as almost constant regardless of external stimuli, resulting in two principal effects [34,35,36,37,38]. First, unlike dissociative CANs, which transform from a solid state more suddenly [39], an extended gummy/rubbery phase can be observed in vitrimers when heated. Second, some researchers have also remarked a greater creep/solvent resistance for vitrimers [37,40,41]. Moreover, vitrimers are distinguished according to their temperature-dependent viscoelastic behavior, as the covalent exchange rate is related to the transition temperature. At high temperatures, when the exchange reactions become fast enough, the viscosity of vitrimers is basically controlled by the exchange reactions, leading to a decrease in viscosity with the temperature that follows the Arrhenius law.

The viscoelastic behavior of vitrimers changes with the topology freezing temperature (T_v_) [28]. The T_v_ is chosen by agreement as the temperature at which the viscosity reaches 10^12^ Pa∙s [42]. Below T_v_, vitrimers behave as conventional thermosets, and above T_v_, they can undergo creeping and relaxing stresses. The control of this temperature is essential since the exchange of covalent bonds and permanent crosslinking allows or does not allow thermal recycling [43,44,45]. Thus, epoxy vitrimers are a substantial advance for the replacement of current thermosets.

The developed vitrimers so far have been overviewed according to the nature of the dynamic exchange reaction. The most common dynamic interactions used in the design of vitrimers have been carboxylate transesterification [28], transamination of vinylogous urethanes [46], transalkylation of triazolium salts [47], disulfide exchange [48] or Schiff base (imine) exchange [49]. Among them, Schiff bases show great potential in the fabrication of epoxy vitrimers due to the presence of a reversible covalent bond since it can be hydrolyzed to aldehyde or ketone and to amine under acid conditions [50]. Furthermore, compounds obtained from natural resources can be used as reagents, which are more attractive and interesting compared with the existing materials. Among them, recent studies show the design of sustainable epoxy vitrimers with natural phenolic compounds such as vanillin (VAN) as the starting material due to the stiffness structure provided by the benzene ring leading to high-T_g_ epoxy vitrimers combined with superior mechanical strength and modulus [51,52,53]. It should be noted that VAN is one of the few compounds with a phenolic group manufactured on an industrial scale from biomass, especially from tannin and lignin [54]. Therefore, this reagent has the potential to become a key precursor for the synthesis of bio-based polymers as it presents an aromatic structure that can achieve good thermo-mechanical properties.

Taking into account all the premises described, in this work, a biovitrimer is developed from components obtained from natural resources, such as epoxidized soybean oil (ESO) and VAN. ESO is used as a bio-based monomer and a vanillin derivative as a new Schiff base to act as a biobased hardener. First, the Schiff base is prepared using VAN and aliphatic diamine (1,4-butandiamine, BDA). Second, ESO and the new Schiff base are combined to form the vitrimer. However, numerous studies have shown the susceptibility of vitrimers to creep substantially under use conditions [55,56,57]. For that reason, in order to improve the properties of this new material, a critical fraction of permanent crosslinks (30, 50 and 70 wt%) is added to the new biovitrimer, which has little or no detrimental effect on reprocessability [48,52,58,59]. During this work, the poly(bisphenol A-co-epichlorohydrin) glycidyl end-capped is used as a commercial epoxy resin (ER). Finally, in all the samples, thermal and mechanical properties, as well as their reprocessability or recyclability, have been investigated.

## 2. Experimental Section

### 2.1. Reagents

Vanillin (VAN, 99%), 1,4-butanediamine (BDA, 99%), 1,2-dimethylimidazole (DMI, 97%) and commercial epoxy resin (ER) poly(bisphenol A-co-epichlorohydrin) glycidyl end-capped (Mn~355 g mol^−1^) were purchased from Sigma Aldrich (Saint Louis, MO, USA). Methanol (MeOH, ≥99.5%) was obtained from PANREAC (Barcelona, Spain). Epoxidized soybean oil (ESO) EPOVINSTAB H-800-D was kindly supplied by Hebron S.A. (Barcelona, Spain). All the chemicals were used as received.

### 2.2. Synthesis of Vanillin-Derived Schiff Base Curing Agent

The vanillin-derived Schiff base (VSB) curing agent was synthesized by dissolving 10.0 g (66 mmol) of vanillin in 100 mL of methanol and mixed in a 500 mL single-necked round-bottomed flask with 3.3 mL (33 mmol) of 1,4-butandiamine. A yellow powdered product (VSB) was obtained (Figure 1a) after solvent evaporation, which was washed with methanol and vacuum dried at 50 °C for 24 h.

### 2.3. Synthesis of Vitrimers

The VSB-cured ESO biovitrimer (VESOV) was synthesized by a two-stage procedure pre- and post-curing. To achieve the pre-curing stage, predetermined amounts of VSB and ESO were added to the reaction using an equivalent phenolic/epoxy hydroxyl ratio (simplified as X). Previous works performed by Zhao et al. [60] and Zeng et al. [61] demonstrated that the optimal ratio to obtain a vitrimer with a highly crosslinked network is X = 0.7. First, the VSB was heated until reaching its melting point in a 100 mL single-necked round-bottomed flask placed into an oil bath and stirred with a magnetic stirring bar under a nitrogen atmosphere. When the VSB was completely melted, the corresponding amount of ESO was added. After ESO and VSB were fully mixed, the catalyst 1,2-dimethylimidazole (DMI) (0.5 wt%) was added, allowing the resulting mixture to react until the magnetic stirrer could not turn due to the increase in the viscosity of the medium (Figure 1b). Then, the post-curing stage started with the transfer of the resultant mixture to a 10 cm × 10 cm × 1.0 mm stainless steel mold, which was placed in a compression molding machine at 150 °C under 10 bar for 2 h. After allowing the sample to reach room temperature, a film was obtained. Figure 1c shows in purple and red how the network structure is reorganized by the thermal-induced exchange reaction of Schiff base in the crosslinking structure of VESOV, leading to the stress relaxation behavior.

For the synthesis of VESOV+ER, the same procedure was used, adding the epoxy resin at the same time as the ESO. Thus, it was decided to add progressive amounts by weight of epoxy resin (30, 50 and 70 wt%) to determine the influence of the addition of epoxy resin to the VESOV. Additionally, to investigate the reaction between VSB+ER, the same procedure as ESO was followed, adding ER to the VSB. Therefore, the mechanical and thermal properties of five samples were studied: VESOV, VESOV+ER (30 wt% of ER), VESOV+ER (50 wt%), VESOV+ER (70 wt%) and VSB+ER.

## 3. Characterization

Fourier transform infrared (FTIR) spectra with wavelengths from 4000 to 400 cm^−1^ were recorded by a Nicolet Nexus spectrophotometer (Thermo Fisher Scientific Inc., Madison, WI, USA), where the samples were measured within KBr pellets. The resolution and scanning number were 4 cm^−1^ and 32 times, respectively. The data were analyzed using OMNIC 8.2 software.

Proton nuclear magnetic resonance (^1^H-NMR) spectra were recorded on a Bruker AV-600 (600 MHz) spectrometer (Billerica, MA, USA) using deuterated chloroform (CDCl_3_) as the solvent.

The melting temperature of VSB and the glass transition temperature of the vitrimers were measured by differential scanning calorimeter (DSC) with a DSC METTLER TOLEDO 822^e^ instrument (Greifensee, Switzerland) equipped with STAR^©^ v14.0 software. The samples (~10 mg) were placed in 100 μL aluminum crucibles. Samples were heated from −10 °C to 250 °C, in the case of the VSB, and from −10 °C to 150 °C for the vitrimers. A scanning rate of 10 °C·min^−1^ under a nitrogen atmosphere with a flow rate of 20 mL·min^−1^.

The thermal stability of the samples (~10 mg) was measured by thermal gravimetric analysis (TGA) under a nitrogen atmosphere (20 mL·min^−1^) with a temperature range of 25–800 °C and a heating rate of 10 °C·min^−1^ by a SHIMADZU DTG-60 thermal gravimetric analyzer (Kyoto, Japan). The statistic heat-resistant index temperature (T_s_) is a characteristic of the thermal stability of the cured resin [62] and was calculated according to Equation (1):(1)$$T_{s}=0.49\times \left[T_{5\%}+0.6\times (T_{30\%}-T_{5\%})\right]$$
where T_5%_ and T_30%_ are the temperatures at, respectively, 5% and 30% weight loss. T_5%_ was considered the onset decomposition temperature (T_o_) of the sample.

Dynamic mechanical analysis (DMA) was carried out in the tensile mode by a DMA1-METTLER TOLEDO instrument (Greifensee, Switzerland) equipped with STAR^©^ v14.0 software for curve analysis. Storage modulus (E′), loss modulus (E″) and loss factors (tan δ) values were collected at 3 °C·min^−1^ heating rate from −10 to 150 °C, displacement of 20 µm, and 1, 3 and 10 Hz frequencies. Rectangular-shaped testing bars with a width of 5.0 mm, length of 10.0 mm and thickness of 0.5 mm were prepared. The glass transition was assigned at the maximum of the loss factor (tan δ = E″/E′).

Reprocessing tests were performed on the compression molding machine (20 TM Hot Plates Press, Hidrotecno S.L., Oiartzun, Spain). The films were cut into small pieces with scissors, placed into the 10 cm × 10 cm × 1.0 mm stainless steel mold, and reprocessed at 150 °C for 60 min at 10 bar. After cooling to room temperature, the reprocessed films were obtained, and their thermo-mechanical properties were measured.

## 4. Results and Discussion

Epoxy thermosets exhibit tremendous thermo-mechanical properties; however, they lack the capacity to be reprocessed after their use. The main objective of this work is to obtain a substitute for epoxy resins that can be considered biobased. In this regard, it must be taken into account that to consider a material as biobased, a minimum of 50% of the compounds incorporated into its formulation must be obtained from natural resources. For this, first, the Schiff base is synthesized using VAN and aliphatic diamine (1,4-butandiamine, BDA). Then, a sustainable epoxy vitrimer derived from vegetable oil ESO is prepared, and finally, to improve the thermo-mechanical properties of the reprocessable vitrimer, a small amount of commercial epoxy resin (ER) is added.

### 4.1. Synthesis and Characterization of Vanillin-Derived Schiff Base (VSB) Hardener

VSB hardener was synthesized by refluxing VAN and BDA in methanol for 24 h. The chemical structure of VSB was confirmed by FTIR and ^1^H-NMR. Figure 1a shows how the characteristic stretching peak of C=O on the aldehyde group (1670 cm^−1^) of the VAN and the broad stretching of N-H (between 3400 and 3250 cm^−1^) of BDA is not observed in the VSB FTIR spectrum. Instead, a new characteristic peak appears at 1655 cm^−1^ for VSB, attributing to the formed Schiff base unit (the most significant FTIR signals were compiled in the Supporting Information). Moreover, ^1^H-NMR easily (Appendix A) allows the Schiff base identification via the disappearance of the signal from the aldehyde group of the VAN and the presence of a new signal in the VSB typical of an imine (Figure 1b). Precisely, the signals for the H proton of alcohol, imine, benzene ring, methoxy and butane were observed at shifts of 9.8, 8.2, 7.4–6.9, 3.9 and 1.8 ppm, respectively. These facts indicate that the Schiff base bond has been successfully formed by the reaction between the amino and aldehyde groups. In addition, the DSC curve of VSB shows only one sharp melting peak at 153 °C (Figure 1c), observing that the VSB was successfully synthesized, as the correspondent melting peak of VAN (~86 °C) was not observed in the DSC curve of VSB. In short, it is corroborated that the vanillin-derived Schiff base was prepared successfully.

### 4.2. Synthesis and Characterization of VESOV, VESOV+ER (Different Percentages) and VSB+ER

Once the VSB was formed (Figure 1a), using DMI as a catalyst, five different formulations of ESO and/or ER vitrimers were developed: VESOV, VESOV+ER (30 wt%), VESOV+ER (50 wt%), VESOV+ER (70 wt%) and VSB+ER. In all cases, the synthetic procedure was the same as described above; after a pre-curing stage where the VSB is reacted with the ESO, the ER or a mixture of both, the post-curing is accomplished (Figure 1b). Subsequently, the thermo-mechanical properties of the samples were studied.

First, the thermal stability of the samples was studied to confirm that the material remains stable and does not suffer any degradation during the reprocessing procedure. The thermal stability profiles of the obtained vitrimers are displayed in Figure 2 and Table 1 and Appendix A. Clearly, all samples are thermally stable up to a temperature of at least 272–297 °C (onset decomposition temperature, T_o_, in Table 1), demonstrating that they possess acceptable thermal stability under all conventional modeling approaches and that they are also thermally resistant during reprocessing, which is important for the recycling of the vitrimer through thermal processing [63].

From Table 1, it can be observed that the statistic heat-resistant index temperature values (T_s_) show the same trend as T_o_ (or T_5%_) values. The results demonstrate that samples with low epoxy resin (ER) content exhibit higher thermal stability (VESOV has the highest), whereas increasing the ER content (VESOV+ER) decreases the thermal stability. The reason for this behavior could be associated with the higher amount of VSB in the network, involving a higher content of imine bonds. Moreover, the higher the oxirane ring content, the more ester and hydroxyl groups are created through the curing. These functions can also promote the thermal scissions of the networks [64].

The glass transition temperature (T_g_), which usually acts as the upper limit use temperature for thermosetting materials, is a major parameter [65]. Figure 3a,b shows the DSC curves of epoxy vitrimers to make a comparison of the T_g-onset_ (simplified as T_g_) between VESOV, VESOV+ER (different percentages) and VSB+ER.

In Figure 3a, it was observed that in the VESOV+ER samples (30 and 50 wt%), two glass transition temperatures coexist. The first one appears at the same range as the T_g_ peak of VESOV, whilst the other T_g_ appears a little more displaced at higher temperatures. This may be due to the formation of two polymeric networks that do not mix with each other; that is, a heterogeneous mixture was obtained. However, as noted below, it is most likely that the post-curing process (2 h at 150 °C) was not enough to obtain a fully cured vitrimer. In Table 2, the T_g_ of the different samples is summarized (T_g-onset_).

As can be noticed in Table 2, when the fraction of epoxy resin increases, the T_g_ values rise significantly; that is, higher stiffness is achieved by making the fluidity of the chains more difficult. Therefore, the addition of a small amount of epoxy resin helps VESOV to improve its thermal properties. Indeed, VESOV has a T_g_ equal to 28.7 °C, while the addition of only 30 wt% of ER increases the T_g_ to 48.2 °C. However, by continuing to add more percentage in weight in ER (50 and 70 wt%), the increase that occurs in T_g_ is not as significant (T_g_ = 48.7 and 50.3 °C for 50 and 70 wt% in ER, respectively). That is, a larger increase in the added proportion of ER did not provide significantly higher T_g_.

Further, the T_g_ of the samples was determined via DMA. Figure 4 shows both the storage modulus (E´) and the loss factor (tan δ) (its maximum peak is the T_g_ of each sample) versus temperature. In addition, Table 2 shows the glass transition temperatures of all the samples obtained by DMA. All the curves with their corresponding numerical data are presented at a frequency of 3 Hz.

As in the DSC, it is observed that while increasing the amount of epoxy resin, the glass transition temperature of the sample increases progressively. On the other hand, in Figure 4a, the tan δ curve of VSB+ER exhibits two peaks, implying it is not completely cured, as not all the chains of the system have reacted effectively. That is, as said before, the post-curing process is not enough to reach a total curing, and samples need a greater post-curing. Furthermore, the difference in the storage modulus is notable amongst samples. Thus, while VESOV has a very low storage modulus, it is remarkably increased by adding a small amount of epoxy resin.

In conclusion, analyzing the results obtained so far, it can be said that adding a small amount of epoxy resin (30 wt%) considerably improves the thermo-mechanical properties of the VESOV vitrimer. Finally, the reprocessability of these vitrimers was studied.

### 4.3. Reprocessability of VESOV, VESOV+ER (Different Percentages) and VSB+ER

The dynamic character of the Schiff base exchanges improves the ability of the samples to be reprocessed. The reprocessing is performed as explained before: The samples were cut into small pieces with scissors, placed into a steel mold and reprocessed at 150 °C for 60 min at 10 bar. These reprocessed samples are shown in Figure 5, observing that VESOV has a considerable ability to be reprocessed (Figure 5a) since it contains dynamic covalent bonds. On the contrary, after carrying out this test with the VSB+ER (Figure 5e), incomplete recovery of the original shape is achieved, as it includes permanent crosslinks due to epoxy resin’s nature as a thermoset. Figure 5b–d also shows the reprocessability of those vitrimers synthesized using different proportions of ESO and epoxy resin (VESOV+ER). In Figure 5b, it can be seen that only with the inclusion of 30 wt% of epoxy resin is it observed that the reprocessability of the sample substantially improves. In order to compare and verify if any degradation occurs after reprocessability, FTIR analysis was performed. In the supporting information, FTIR spectra of all the samples are included. No differences were observed between pre- and post-reprocessed spectra (Appendix A).

Yet, it is necessary to verify that this reprocessing does not diminish the good thermo-mechanical properties obtained for the original samples. For this, the characterizations via DSC and DMA of the different reprocessed samples were accomplished. Moreover, a comparison between the original and remolded samples is performed to evaluate how many times samples can be reprocessed without losing their thermo-mechanical properties.

In DSC, the reprocessed samples only present a T_g_, showing that they are totally cured (Figure 3b). In addition, as seen in Table 2, all the glass transition temperatures obtained for the reprocessed samples are similar or even higher than those corresponding to the samples without reprocessing. The only exception is precisely the vitrimer synthesized only with ESO (VESOV), which showed a clear decrease in its T_g_.

However, DMA measurements (Figure 4b) demonstrate that all the reprocessed samples slightly increase their glass transition temperatures (Table 2) because they undergo a post-curing process in which the crosslinking of their bonds increases. This fact is confirmed by the width of the tan δ peaks, as in the original samples, the width of the peak is greater, denoting that the samples are more heterogeneous. In the reprocessed samples, peak width decreases considerably, denoting that the crosslinking of the bonds has been superior and obtained a larger homogeneity of the system.

## 5. Conclusions

In conclusion, it has been possible to verify that 1,4-butandiamine is a suitable reagent to synthesize a vanillin-derived Schiff base (VSB) that can be used as a curing agent in the development of a new vitrimer from epoxidized soybean oil (VESOV). Furthermore, the inclusion of a commercial epoxy resin to the VESOV helped to improve its mechanical and thermal properties, observing by DSC and DMA techniques that when the percentage of epoxy resin increases, the T_g_ increases remarkably. However, these new epoxy materials must have a balance between their thermo-mechanical properties and their reprocessability to be considered vitrimers. In this way, a circular economy can be established, such as thermoplastics. Standing on this, it can be concluded that the VESOV+ER 30 wt% sample seems to be the most suitable to achieve this objective since it has the qualities of a thermoset (high storage modulus and T_g_) and contains elements of a thermoplastic (good reprocessability), with this last property not attributable to conventional epoxy resins.

In short, this new dynamic hardener based on vanillin and 1,4-butandiamine obtained good properties as expected; a sustainable vitrimer was synthesize based only on epoxidized vegetable oil. The first approach to this objective has been made by developing a new material with remarkable features by adding a small amount of commercial epoxy resin (30 wt%), so the material can be considered biobased with good mechanical properties and the possibility of recycling. Therefore, this material can be postulated in the future, with slight improvements, to be a replacement for current commercial epoxy resins.

## Data Availability

Not applicable.

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
