# Peer review of "Improving the Recyclability of an Epoxy Resin through the Addition of New Biobased Vitrimer"

_polymers, 2023, doi:10.3390/polym15183737_

Round 1

Reviewer 1 Report

Referring to modern topic of reproducibility, circular economy and bioproducts in this paper there are important statement of "improving the recyclability..." not simple causing or achieving recyclablity. Recyclability cannot be related to the reduction of mechanical properties of Epoxy based Resins. Epovy based products like boats, coatings, even planes which are designed for a long time service and adding to its properties term "recyclability" cannot mean reduction of mechanical properties after some period of time (designed service life). This point is clearly presented by the Authors in lines 29 - 32.

In this paper there is presented an interesting idea of "net in the net" - CAN - see lines 58 - 60 and vitrimers (line 70).

Authors presented good description and chemical characterization (FTIR, NMR) of crosslinking agent based on Schff base of 1,4 BDA and Vanilline. There are missing more detailed mechanical characterization of resins and resins mixed with 30%; 50% and 70% w/w of commercial epoxies and changes od mechanical properties after extended aging time. I suggest to use for this purpose (accelerated aging) the Pressure Cooker Test conditions. But this can be a main topic for the next paper. In this paper there is enough, especially, that the Authors made by themselves an important statement in lines 341 - 344.

Paper is clearly presented, there are no present editorial mistakes. I would like to ask the Authors to specify tested as additive commercial Epoxy Resin and add its mechanical properties.

Reviewer 2 Report

General comments
Well researched and well written manuscript. Needs minor improvements

Specific comments
Introduction though bit lengthy is well structured and well written.
In line #66 authors use "their reticulated structure.." but doesn't reticulated structures refer to porous materials like MOFs, ZIFs etc?
There are some grammatical mistakes in lines 70 to 83. Rectify.
Line 138: Mistake in the statement "Zhang et al. [59] and Zeng et al. [60].."

More work about re-processability must be reported:
FTIR of pre- and post reprocessed should be included in revised manuscript. This is critical since the photographs exhibited in Fig 5 shows that in none of the cases 'clean films' were formed and there was difficulty in re-molding indicating possible degradation.

Minor grammatical mistakes
